# A Comprehensive Review on the Interplay between *Neisseria* spp. and Host Sphingolipid Metabolites

**DOI:** 10.3390/cells10113201

**Published:** 2021-11-17

**Authors:** Simon Peters, Ingo Fohmann, Thomas Rudel, Alexandra Schubert-Unkmeir

**Affiliations:** 1Institute for Hygiene and Microbiology, University of Wuerzburg, 97080 Wuerzburg, Germany; simon.peters@hygiene.uni-wuerzburg.de (S.P.); ingo.fohmann@uni-wuerzburg.de (I.F.); 2Chair of Microbiology, University of Wuerzburg, 97080 Wuerzburg, Germany; thomas.rudel@biozentrum.uni-wuerzburg.de

**Keywords:** sphingolipids, *Neisseria meningitidis*, *Neisseria gonorrhoeae*, host–pathogen interaction

## Abstract

Sphingolipids represent a class of structural related lipids involved in membrane biology and various cellular processes including cell growth, apoptosis, inflammation and migration. Over the past decade, sphingolipids have become the focus of intensive studies regarding their involvement in infectious diseases. Pathogens can manipulate the sphingolipid metabolism resulting in cell membrane reorganization and receptor recruitment to facilitate their entry. They may recruit specific host sphingolipid metabolites to establish a favorable niche for intracellular survival and proliferation. In contrast, some sphingolipid metabolites can also act as a first line defense against bacteria based on their antimicrobial activity. In this review, we will focus on the strategies employed by pathogenic *Neisseria* spp. to modulate the sphingolipid metabolism and hijack the sphingolipid balance in the host to promote cellular colonization, invasion and intracellular survival. Novel techniques and innovative approaches will be highlighted that allow imaging of sphingolipid derivatives in the host cell as well as in the pathogen.

## 1. Introduction

Sphingolipids form a diverse group of structural related lipids, composed of a sphingoid backbone (sphingosine, dihydrosphingosine (dhSph) or phytosphingosine) coupled via an amid bond to a fatty acid side chain of various length. The great diversity of sphingolipids is achieved through numerous different head moieties ranging from two hydroxyl groups (ceramide) to highly complex sugar structures (glycosphingolipids, e.g., GM1) [1,2]. The physiological functions of sphingolipids are as diverse as their structure and include cell differentiation, growth and death, as well as senescence and inflammatory responses [2]. Sphingolipids are essential for human cells and can be found in every cell type and in various compositions. Therefore, it is not surprising that sphingolipids, and enzymes involved in the sphingolipid pathway, are hijacked by different pathogens including bacteria (e.g., *Neisseria meningitidis*, *N. gonorrhoeae*, *Chlamydia* spp., *Pseudomonas aeruginosa*, *Staphylococcus aureus* or Salmonella), viruses (e.g., rhinovirus, measles and Ebola virus) and parasites (e.g., *Trypanosoma cruzi*) during various steps of their infection process (reviewed in [3,4,5,6,7]).

De novo synthesis of sphingolipids begins in the endoplasmic reticulum with the condensation of palmitoyl-CoA and L-serine, which is catalyzed by the serine palmitoyltransferase (SPT). SPT generates 3-keto-dihydrosphingosine, which is then reduced to dihydrosphingosine (dhSph). The N-linked addition of a fatty acid to dhSph, catalyzed by dihydroceramide synthase, results in generation of dihydroceramide, which is subsequently desaturated to ceramide in mammalian cells. A large number of complex sphingolipids are then derived from ceramides: (i) addition of sugar groups to the ceramide backbone results in formation of complex glycosphingolipids, including cerebrosides, and (ii) further addition of sialic acids leads to the formation of gangliosides. The transfer of a phosphocholine headgroup from phosphatidylcholine to ceramide finally results generation of sphingomyelin (SM). SM deacylase then generates sphingosylphosphorylcholine (Lysosphingomyelin). On the other hand, ceramides can be generated by the action of different sphingomyelinases, that catalyze the hydrolysis of SM into ceramide and phosphorylcholine. Ceramide can be further metabolized to ceramide-1 phosphate by ceramide kinases. The reverse reaction is catalyzed by ceramide-1-phosphate phosphatase. Otherwise, ceramide can be converted to sphingosine, which can be phosphorylated by sphingosine kinases to sphingosine-1-phosphate (S1P). The reverse reaction is catalyzed by S1P phosphatases, or it can be irreversibly degraded by S1P lyase to ethanolamine phosphate and hexadecenal. 

Numerous enzymes of the sphingolipid metabolism can be regulated in response to extra- and intracellular stimuli and in turn serve as regulators of sphingolipid levels. During the last decade some of these enzymes have been identified and described as targets, which are attacked by bacteria or viruses in various ways, in particular the acid and/or neutral sphingomyelinases [8,9,10,11,12,13,14,15,16] and the sphingosine kinases [15,17,18,19,20,21]. In this review we will highlight the impact of the pathogenic *Neisseria* spp., *N. meningitidis* and *N. gonorrhoeae* on those enzymes (Figure 1).

## 2. Sphingomyelinases 

Sphingomyelinases (SMases) are classified into five types based on pH optimum, subcellular localization and cation dependence, including the lysosomal acid sphingomyelinase (L-aSMase), secretory acid sphingomyelinase (S-aSMase), alkaline sphingomyelinase (alk-SMase), Mg-dependent neutral sphingomyelinase (nSMase) and Mg-independent nSMase [22,23]. Goni and Alonso added the group of bacterial SMases as a further sixth type to this classification [24].

## 3. Acid Sphingomyelinase

The acid SMase (aSMase) is one of the best-characterized SMases with an optimal pH of 5 and is either entering the lysosomal trafficking pathway or is constitutively secreted from the Golgi complex to the extracellular space, via the default secretory pathway [22,25]. The enzyme functions predominantly in the degradation of SM, primarily in the lysosomes. Noteworthy, the L-aSMase also shows extracellular activity on the outer leaflet of the cell. The activity in the extracellular space, with its neutral pH, is achieved through an acidic microenvironment created by fusion of the lysosome and outer membrane and the release of V1 H(+)-ATPases [26]. In contrast, the activity of the S-aSMase at the neutral pH of the blood or CSF remains a point of discussion [25]. A genetic deficiency of lysosomal aSMase results in Niemann–Pick disease type A and B [27], an autosomal-recessive lipid storage disorder accompanied by massive accumulation of SM in lysosomes. 

The enzyme was first isolated from urine as a monomeric 72-kDa glycoprotein with a 61-kDa polypeptide core. ASMases are encoded by the SM phosphodiesterase 1 (SMPD1) gene located on the short arm of chromosome 11. Notably, aSMase exists in two forms: a zinc (Zn^2+^)-independent L-aSMase and a Zn^2+^-dependent secreted S-aSMase that arise from alternative trafficking of a single protein precursor. Currently more than 100 missense mutations in SMPD1 are listed in the human Gene Mutation Database; however, not every sequence variation in SMPD1 is detrimental. 

ASMase consists of an N-terminal saposin domain, a proline-rich connector and a C-terminal catalytic domain [28]. The saposin domain of aSMase likely plays an important role in orienting the active site of the catalytic domain towards the membrane [28]. Notably, the crystal structure of aSMase was dissolved in 2016 [29]. 

Activation of aSMase can be initiated in several ways. For example, UV radiation leads to aSMase phosphorylation at serine residue 508 by protein kinase C-delta (PKCδ,) resulting in its activation [30,31]. Other described pathways for aSMase activation include acidification of endosomal compartments, proteolytic cleavage of the N-terminus or regulation of activity via oxidation (e.g., by reactive oxygen species (ROS)) of a C-terminal cysteine residue [32,33,34,35,36]. In addition, diacylglycerol (DAG), which is formed by the hydrolysis of phosphatidylcholine via phosphatidylcholine-specific phospholipase C, is known to be involved in the activation of aSMase by activating PKCδ [30,37]. 

A prerequisite for the activity of L-aSMase at the outer leaflet of the plasma membrane is its translocation from the lysosome the outer membrane. The fusion of membrane proximal lysosomes is known to appear in a calcium-dependent manner as part of cellular repair mechanisms after plasma membrane injury [38,39]. 

## 4. Neutral Sphingomyelinase

Four mammalian neutral SMases (nSMases) have been identified so far. Just like the aSMases, nSMases function in the hydrolysis of SM to generate ceramide and phosphorylcholine. NSMase2 is currently the best studied isoform and is involved in bone mineralization, exosome formation and cellular stress responses [40,41,42,43,44,45,46,47]. In this review we will focus on nSMase2 which plays a key role in the pathogenesis of *N. gonorrhoeae* infection of epithelial cells and will refer to excellent reviews that highlight the biochemical properties and roles of the other nSMases [48,49]. 

The human nSMase2 was identified and cloned in 2000 by Hofmann and colleagues [50]. The activity of nSMase2 has an optimum pH of 7.4, depends on Mg^2+^ ions, and nSMases are presumed to be activated by unsaturated fatty acids and several anionic phospholipids (APL) [50]. NSMase2 has two major functional domains: a hydrophobic N-terminal domain (NTD) associated with the membrane, and a C-terminal catalytic domain (CAT), which is cytosolic. 

The NTD is an integral-membrane domain and contains two hydrophobic segments, which are predicted to be helical but do not span the membrane [51]. The activation mechanism of nSMase2 is still not fully understood; in vitro data, however, indicate that several APLs, including phospatidylserine, phospatidylglycerol and cardiolipin are required for full activity [50,52,53]. 

Notably, nSMase2 harbors two palmitoylation sites that are important for correct subcellular localization and contribute to nSMase2 membrane association [51]. NSMase2 is localized to the plasma membrane, Golgi and recycling compartments. Upon treatment with TNF-α, phorbol 12-myristate 13-acetate (PMA) or hydrogen peroxide, nSMase2 translocates from the Golgi to the plasma membrane. This translocation may regulate activity through the access to SM or activating APLs. 

## 5. Sphingosine Kinases

Sphingosine kinases (SphKs) catalyze the phosphorylation of sphingosine to generate S1P. The Spinster-2 transporter secretes S1P from the cytosol into the extracellular space where it binds to the surface G-protein-coupled receptors (S1PR1-5) which modulate a large number of different cellular processes, such as embryonic angiogenesis, lymphocyte recirculation, heart rate and brain endothelial permeability [54,55,56,57,58,59]. Two different SphK isoforms are known to date (reviewed in [60]). 

SphK1 was first identified in mice and rats as a mainly cytosolic, Mg^2+^-dependent enzyme with a broad pH optimum between pH 6.6 and 7.7 [61,62]. Human SphK1 can be differentiated into three splice variants, hSphK1a, hSphK1b and hSphK1c with a respective size of 384 amino acids (aa), 398aa and 470aa and comparable kinetics but differing tissue distribution [63,64,65]. Several mechanisms induce SphK1 translocation from cytosol to the plasma membrane, e.g., extracellular-related-Kinase (ERK) 1/2-mediated phosphorylation of Ser225, phosphatidyl serine interaction, Fyn/Lyn kinase interaction or Ca^2+^ signaling via calcium- and integrin-binding protein 1 bringing SphK1 in close contact with its substrate sphingosine and thereby increasing S1P production [66,67,68,69,70,71,72]. Recent studies also suggest that translocation might be dependent on SphK1 dimerization [73]. Additionally, the before-mentioned modifications and interactions may directly influence SphK1 activity through conformational changes, which are now verifiable, since SphK1 crystal structure was resolved by Wang and colleagues to allow modeling of the ATP- and the sphingosine-binding domains [74]. Besides Spinster-2-transporter-mediated export of S1P, SphK1 itself is also secreted to the extracellular space, where it contributes to plasma-vasculature S1P gradient [58,63,75,76,77]. 

SphK2 is predominantly found in the nucleus because of its N-terminal nucleus localization sequence, where it influences DNA synthesis and causes cell cycle arrest [78,79,80]. However, SphK2 can also be exported upon protein kinase D-induced phosphorylation [81]. SphK2 appears in three splice variants, namely SphK2a, SphK2b and SphK2c with a size of 618aa, 654aa and 716aa, respectively [82,83]. SphK2b is the main variant, with a 1.3-fold higher activity compared to SphK2a [64,79]. SphK2 has a broader substrate specificity than SphK1 but favors dhSph over sphingosine [84]. In contrast to SphK1, SphK2 shows pro-apoptotic features dependent on its subcellular localization, e.g., through increase of ceramide levels at the endoplasmic reticulum or Bcl2-homologous-antagonist/killer (BAK)-induced cytochrome c release from mitochondria [85,86,87]. On the other hand, mitochondrial S1P production by SphK2 is essential for cytochrome c oxidase assembly and ERK1/2-mediated phosphorylation of cytosolic or plasma-membrane-associated SphK2 promotes survival [88,89,90]. The latter might reflect the fact that both kinases can potentially compensate for each other [91,92].

Several studies have demonstrated that SphKs and their product, S1P, enhance the replication of various viruses, including influenza, measles, and hepatitis B virus (reviewed in [93]). Moreover, SphK-mediated S1P production is also targeted by bacteria, as shown for *P. aeruginosa*, *Escherichia coli*, *Streptococcus pneumoniae* and *N. gonorrhoeae* during their interaction with host cells [15,17,18,19]. Recent findings from our group also indicate a transient activation of SphKs during *N. meningitidis* infection (unpublished data). Here, we will highlight the role of SphKs in the infection process of *N. gonorrhoeae*. 

## 6. *Neisseria* spp.

The genus *Neisseria* consists of a large group of Gram-negative betaproteobacteria with approximately 30 different species [94]. Whereas the majority of the genus is considered as non-pathogenic human colonizers, two members are associated with pathological infections: *N. meningitidis*, the major cause of bacterial septicemia and meningitis worldwide and *N. gonorrhoeae*, the etiological agent of gonorrhea, a sexually transmitted infection (STI) [95,96]. 

## 7. Role of Sphingolipids in *N. meningitidis* Pathobiology

*N. meningitidis* transmission from a host to another occurs by aerosols and is followed by the colonization of the nasopharynx, where *N. meningitidis* resides as a non-pathogenic commensal in 10–40% of the healthy population [97,98]. In rare cases, and by still unknown mechanisms, *N. meningitidis* is able to overcome the epithelial barrier and reaches the bloodstream. Once in the bloodstream, *N. meningitidis* proliferates and may cause severe vessel damage (pupura fulminans) and, after crossing the endothelium of the blood–cerebrospinal fluid barrier (BCSFB), infects the meninges [99]. Although potent vaccines are available, *N. meningitidis* is still the leading cause of bacterial meningitis in toddlers and young adults with 500,000 to 1.2 million infections per year and a case lethality rate between 10% and 15% [100,101]. 

A crucial step in the pathogenesis of meningococcal meningitis is the disturbance of the cerebral microvascular endothelial function, resulting in breakdown of the BCSFB. Endothelial cells (ECs) of the BCSFB are characterized by strongly regulated endocytosis and enhanced cell–cell contacts, mediated by specialized tight junction proteins [102]. The outer membrane protein OpcA was identified as a potential contributor to *N. meningitidis* invasion into brain ECs (BECs) [103,104]. It confers a tight association of the bacterium to extracellular matrix proteins, such as vitronectin and fibronectin and OpcA interaction with these factors leads to binding to endothelial αVβ3 integrin and/or α5β1-integrin on BECs [103]. This interaction triggers cytoskeletal rearrangements and uptake of meningococci [105,106,107]. In addition to integrins, heparan sulphate proteoglycans (HSPGs) act as cell receptors for the meningococcal OpcA invasin [108]. HSPGs are known to activate the phosphatidylcholine-specific phospholipase C, leading to the release of diacylglycerol, a well-characterized lipid second-messenger. 

As mentioned in the introduction diacylglycerols can activate aSMases and interestingly this pathway is triggered after meningococcal OpcA binding to HSPGs, finally resulting in increase of ceramide on the host cell surface (Figure 2 and Figure 3) [8,109]. Ceramides in the outer leaflet of the plasma membrane then condense into so-called ceramide-enriched membrane platforms (CRPs) [110]. CRPs offer a unique membrane composition, and it was shown that the tyrosine kinase receptor ErbB2, important for meningococcal invasion, clusters in these platforms [10]. 

Of note, cell culture-based experiments revealing the capability of *N. meningitidis* to activate aSMase suggested a correlation with epidemiological data and clinical presentation. Multilocus sequence typing (MLST) analysis classifies meningococcal isolates according to polymorphisms in seven housekeeping genes. Analysis of meningococcal population by MLST suggests that disease-causing meningococci belong to particular groups of related sequence types (STs), termed clonal complexes (CCs), particular CCs that are overrepresented in disease isolates relative to their carriage prevalence and only few so-called hyperinvasive lineages are responsible for most reported diseases [97,111]. Cell culture-based data showed that isolates of the ST-11 cc (lacking the *opcA* gene) barely induced aSMase activation and ceramide release, correlating with significantly lower bacterial uptake by BECs [10]. Of note, isolates of the ST-11 cc are reported to cause severe sepsis, but rarely meningitis.

Besides OpcA, type 4 pili (T4P) also contribute to aSMase translocation, from membrane proximal lysosomes, and CRP formation [11] (Figure 3). T4P mediates the initial contact of *N. meningitidis* to eukaryotic cell surfaces, and are involved in bacterial movement, also known as ‘twitching motility’, and transformation competence. Moreover, T4P trigger calcium release and stress kinase signaling in the infected eukaryotic cell as well as cytoskeletal rearrangements, and ‘cortical plaque’ formation [112,113]. They are composed of a main component, the major pilin PilE, that assembles into a helical fiber. The helical assembly of pilin into fibers relies on proteins located in or in the vicinity of the cytoplasmatic membrane [114]. In 2014 Bernard et al. showed that *N. meningitidis* utilizes CD147, a member of the immunoglobulin superfamily, for T4P-dependent adhesion to ECs and demonstrated the central role of CD147 for vascular colonization of pathogenic meningococci [115]. Noteworthy, we demonstrated that interaction between T4P and CD147 led to a calcium-dependent translocation of the L-aSMase from the lysosome to the outer leaflet of the plasma membrane and its activation, followed by subsequent ceramide release on BECs [11,115]. The increase of intracellular free calcium occurred through the activation of inositol 1,4,5-trisphosphate receptors on the endoplasmic reticulum [116]. This finding was observed after infecting cells with living bacteria, as well as treatment of the cells with pilus-enriched fractions [11] (Figure 3). The rise in intracellular free calcium triggers cellular repair mechanisms in which membrane proximal lysosomes starts to fuse with the membrane and thereby deliver aSMase to the surface [38]. 

Glycosphingolipids (GSLs) represent a subgroup of sphingolipids known for their great variability and high abundance. GSLs, which are localized with surface-exposed oligosaccharides in the outer leaflet of the plasma membrane, have been shown to act as specific entry platforms for bacteria and their toxins such as Shiga toxin, cholera toxin and other related AB5-subunit toxins [117,118,119]. We recently revealed for the first time that the GSL monosialotetrahexosylganglioside (GM1) among the several gangliosides may serve as a receptor for *N. meningitidis* to invade the endothelium of the BCSFB [120]. Experiments with the two native sphingolipids globotriaosylceramide (Gb3) and raft-associated GM1, respectively, in bacterial invasion demonstrated that GM1 labeled by Alexa Fluor 647 Cholera toxin B accumulated strongly around meningococci and thus support infection [120]. In addition, inhibition of the meningococcal–GM1 interaction by treatment with Cholera toxin B significantly reduced the ability of *N. meningitidis* to invade the host cell [120]. However, the exact mechanism how *N. meningitidis* targets GSLs remains elusive.

## 8. Role of Sphingolipids in *N. gonorrhoeae* Pathobiology

*N. gonorrhoeae* is the causative agent of gonorrhea, a STI, representing a major healthcare problem worldwide with nearly 80 million cases per year [95]. Pathology in females occur in approximately 10–20% of the infected individuals and manifests primarily through the activation of the host innate immune system leading to pelvic inflammatory disease (PID). PID increases the risk for ectopic pregnancy, tubal-factor infertility and chronic pelvic pain [121]. In contrast to females, infected males are often symptomatic and display typically purulent penile discharge and dysuria [122]. One of the main concerns of *N. gonorrhoeae* infection is the development of strains resistant against almost all classes of antibiotics including expanded-spectrum cephalosporins, while an effective vaccine is lacking. Therefore, new treatment options are needed as well as an increased investment into *N. gonorrhoeae*-specific drug discovery [123].

The first study to demonstrate that bacteria are able to activate SMases was conducted by Grassmé and colleagues in 1997 [8]. They showed that *N. gonorrhoeae* is able to increase aSMase activity and induce ceramide formation in non-phagocytic RT-112 human bladder tumor cells [8]. Similar as shown for *N. meningitidis*, *N. gonorrhoeae* triggers aSMase activation via interaction of an outer membrane protein (here Opa50), with HSPG receptors of the syndecan family. In line with the data published for *N. meningitidis*, the phosphatidylcholine-specific phospholipase C is activated after Opa50/HSPG interaction followed by release of diacylglycerol, that finally leads to aSMase activation [8,10] (Figure 3). Moreover, *N. gonorrhoeae* is capable to trigger aSMase activation downstream of Opa52/CEACAM interaction in professional phagocytes [9] (Figure 3). In phagocytes CRPs are linked to an increase in intracellular killing in the phagosome, as the fusion of ceramide-containing phagosomes with late endosomes is induced [124].

If left untreated, *N. gonorrhoeae* can spread through the bloodstream and lead to disseminated infection [125]. Disseminated gonococcal infections (DGI) occur in approximately 0.5 to 3 percent of adult patients infected with *N. gonorrhoeae*. The symptoms of DGI range from mild joint problems or isolated skin lesions without fever to polyarthritis and high fever. Patients with DGI usually have no urogenital symptoms. Epidemiological studies show a connection between DGI and the expression of PorB porin of serotype A (PorB_IA_). The expression of serotype B gonococcal porin (PorB_IB_), on the other hand, is associated with the development of the local infection. PorB_IA_ mediates the rapid invasion under phosphate-free conditions into the epithelial cell, which takes place via the binding to the scavenger receptor expressed on endothelial cells (SREC-I) [12,126]. Interestingly, PorB_IA_/SREC-I-initiated internalization triggers nSMase activation, in particular nSMase2, whereas aSMase does not seem to play a role in the PorB_IA_-mediated invasion [12]. However, the exact mechanism by which nSMase2 is activated is unknown. As a consequence of nSMase2 activation in response to PorB_IA_/SREC-I interaction, CRPs are formed and allow the spatial organization of downstream signaling molecules such as phosphoinositol-3 (PI3) kinase [12]. 

Intriguingly, a recent published study indicated an important role of sphingosine in the intracellular survival of *N. gonorrhoeae*, leading to the assumption that sphingosine might be a crucial metabolite for the host cell to regulate the infection with intracellular pathogens [18]. The antibacterial activity of sphingosine on numerous bacterial species or fungi has already been described [127,128,129] and will be highlighted in a following paragraph. Solger et al. demonstrated that the increase of intracellular sphingosine levels due to inhibition of sphingosine kinases by various pharmacological inhibitors significantly impaired the survival of intracellular gonococci [18]. By using a recently developed clickable azido-sphingosine this sphingolipid could be co-localized by confocal immunofluorescence microscopy as well as expansion microscopy on the surface of intracellular gonococci [18]. 

## 9. Visualization of Sphingolipids in *Neisseria* spp.

Studying the interactions between (sphingo-)lipids and bacteria remains challenging and very few tools allow for detection of (sphingo-)lipids in bacteria. Traditionally biochemical, biophysical and structural methods are used to study e.g., lipid–lipid or lipid–protein interactions in membranes [130]. Moreover, lipidomic approaches offer the opportunity to quantify lipids extracted from tissues or even different cell organelles to yield novel information of the global lipid composition under a variety of (patho)physiological conditions [131]. Advances to fluorescence microscopy opens up new avenues for nanoscale imaging and tracking of lipids in living cells. Synthetic analogues such as borondipyrromethene (BODIPY) sphingosine or 6-([N-(7-nitrobenz-2-oxa-1,3-dimensionaliazol-4-yl)amino]hexanoyl) sphingosine (NBD-Cer) are commercially available and can be used in sphingolipid transport and metabolism studies; however, these probes display huge fluorophores in their fatty acid or sphingosine chains that may perturb membrane architecture. Ceramide-specific antibodies have been successfully generated, though are mainly applied on fixed cells to avoid artificial clustering of target molecules [110,132]. 

To overcome these limitations, ‘clickable’ sphingolipids with reactive groups have been developed [133,134,135,136,137,138]. These groups react spontaneously with either an alkyne or a tetrazine in a ‘click-chemistry’ reaction [139]. This novel tool has now successfully first been applied to visualize uptake and localization of sphingolipids in mammalian cells [134,135]. For example, azido-functionalized ceramides were incorporated in membranes of activated living T cells [134]. These and further functionalized sphingolipids were also successfully applied to *N. meningitidis* and *N. gonorrhoeae* [18,133,137,140], where they showed to exhibit bactericidal effects (outlined in the following paragraph). Confocal laser scanning microscopy (CLSM) and imaging by direct stochastic optical reconstruction microscopy (*d*STORM) demonstrated homogeneous distribution of the clickable azido-modified ceramide analogue in the bacterial membrane of *N. meningitidis* [133]. The clickable azido-modified sphingosine could be co-localized by confocal immunofluorescence microscopy as well as expansion microscopy on the surface of intracellular gonococci [18], further corroborating the suggestion that the increase of intracellular sphingosine decreases intracellular survival of the bacteria. 

## 10. Antimicrobial Activity of Sphingolipids against Pathogenic *Neisseria* spp.

In addition to the involvement of eukaryotic sphingolipids and their related metabolic enzymes in the pathogenicity of *Neisseria* spp., another mode of interaction has been uncovered in recent years. A subset of sphingolipid metabolites has been shown to have growth inhibitory effects on bacteria as well as fungi [127,141,142,143,144]. The inhibition of growth depends on the structure of the sphingolipid as well as on the microorganism. So far, sphingosine in particular shows a promising antibacterial effect. Fischer et al. determined the antimicrobial activity for sphingosine against *E. coli, S. aureus* and *Corynebacterium* spp. with minimal inhibitor concentrations of 1.3 to 7.8 µg/mL [144,145]. Of note, inhaling sphingosine in cystic fibrosis patients prevents respiratory infections and can even cure an existing *P. aeruginosa* infection [146]. Furthermore, dihydrosphingosine shows a good effectiveness against multiresistant *Mycobacterium tuberculosis* isolates [147]. Synthetic analogues of dihydrosphingosine, which are also effective against *M. tuberculosis*, were presented in 2009 [147].

An effective antibacterial activity of sphingosine against pathogenic *Neisseria* spp. was first demonstrated for *N. meningitidis* by Bibel et al. in 1993 [127]. In addition, recently published data demonstrated that short chain ceramides (C6-ceramide) and the synthetic azido-modified analogue of C6-ceramide (ω-azido-C6-ceramide), initially developed for monitoring sphingolipid dynamic in living mammalian cells as outlined above, also exhibited an effective antibacterial activity against *N. meningitidis* and *N. gonorrhoeae* [133]. Importantly, the compound showed no cytotoxic effects on eukaryotic cells [133]. 

However, although the antimicrobial activity is known for a long time, the underling mechanism of the growth inhibitory effect remains elusive. Recently, one publication pointed towards a possible mechanism of the antimicrobial effect of sphingosine against *P. aeruginosa* and *E. coli*. The authors demonstrated that bacterial membrane disruption is induced by the interaction of the protonated primary amid of sphingosine with the highly negatively charged cardiolipin in the plasma membrane of the bacteria [148]. This finding provides a plausible explanation for the effect observed for sphingosine, but cannot explain the antibacterial effect of short-chain ceramides and their analogues, where the primary amid is reduced to a secondary amid for the coupling of the fatty acid side chain [133,137].

Of note, the native or an azido-modified analogue of sphingosine displayed an antimicrobial activity against *Neisseria* spp. even stronger than against *E. coli* and *S. aureus* [18,133,137], whereas the synthetic analogue of C6-ceramide was completely inactive against *E. coli* and *S. aureus* [133,137]. 

With the azido-modified ceramide and the azido-modified sphingosine, drug candidates had been developed which also made it possible to visually follow and represent their incorporation in bacteria with the aid of correlative light and electron microscopy. Pre-embedding labeling used during this application added another level of versatility to the super-resolution array tomography workflow. It enabled super-resolved localization of incorporated fluorescent tags in the full ultrastructural background without on-section labelling [137]. This novel approach allowed to clearly localize the analogues being incorporated in the outer membrane, resulting in its detachment from the inner membrane and rupture. These findings could be substantiated by mass spectrometric analyses after separation of both membranes [137]. 

## 11. Conclusions and Outlook 

Sphingolipids display a tremendous structural variety, which is reflected by their diverse involvement in cellular processes. Therefore, it is not surprising that pathogens developed strategies to use this pathway for their advantages. This fact has been acknowledged by the scientific community during the past decade, and the study of the interaction between pathogens and host cell derived sphingolipids has become a focus of research in infection biology. The development and combination of modern analytical techniques, including mass spectrometry, super-resolution fluorescence microscopy and biorthogonal functional chemistry significantly improve our tools to investigate and understand the interactions between pathogens and host host-derived sphingolipids. 

Understanding the mechanisms of how pathogenic *Neisseria* spp. interact with the host cell sphingolipid metabolism, may allow for identification of novel targets to treat, or prevent infections caused by this pathogen. Of importance, the antimicrobial activity observed for a subset of sphingolipids—and their azido-functionalized analogues—against pathogenic *Neisseria* spp. may lead to the development of novel, sphingolipid-based, antimicrobial compounds.

## Figures and Tables

**Figure 1 cells-10-03201-f001:**
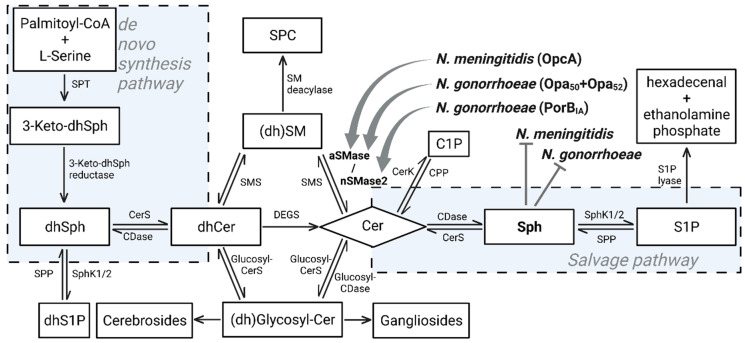
Schematic overview of the sphingolipid metabolic pathway. Both, *N. meningitidis* and *N. gonorrhoeae* are able to modulate sphingomyelinase (SMase) activity in host cells, either the acid SMase (aSMase) or neutral SMase 2 (nSMase2). Intriguingly, high sphingosine levels interfere with growth of intracellular gonococci or exhibit growth inhibitory effects on *N. meningitidis* under in vitro conditions. SPT: Serine palmitoyl transferase; (dh)Sph: (diydro)sphingosine; CerS: Ceramide synthase; CDase: Ceramidase; dhCer: dihydroceramide; DEGS: dihydroceramide desaturase; SMS: sphingomyelin synthase; (dh)SM: (dihydro)sphingomyelin; SPC: sphingosylphosphorylcholine; CerK: ceramide kinase; C1P: ceramide 1-phosphate; CPP: ceramide 1-phosphate phosphatase; SphK: sphingosine kinase; S1P: sphingosine 1-phosphate; SPP: S1P phosphatase; aSMase: acid sphingmyelinase; nSMase2: neutral sphingomyelinase 2.

**Figure 2 cells-10-03201-f002:**
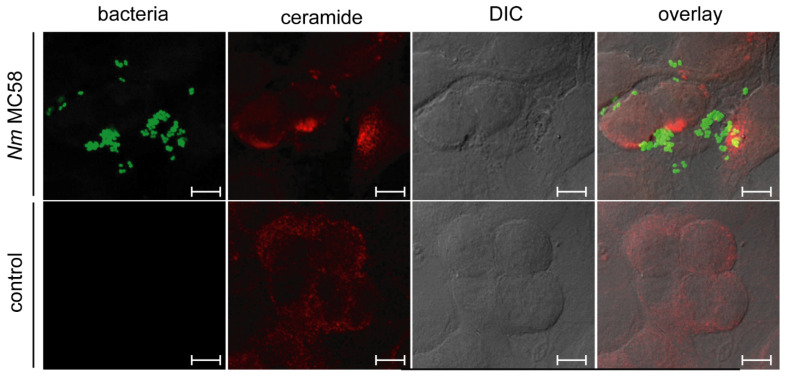
Example image showing ceramide staining in *N. meningitidis* infected brain endothelial cells [10] (copyright permission from the journal).

**Figure 3 cells-10-03201-f003:**
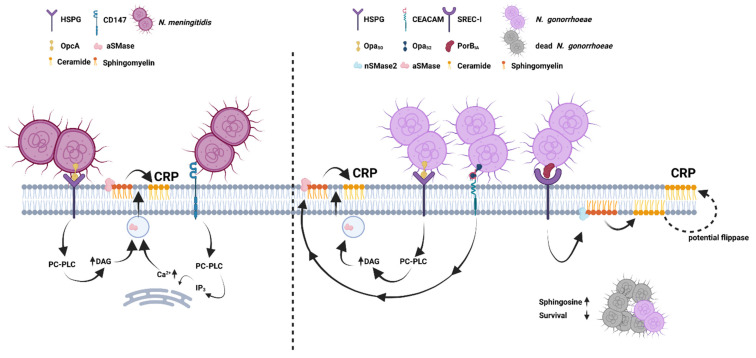
Schematic overview of interactions of *N. meningitidis* (**left** panel) or *N. gonorrhoeae* (**right** panel) with host cells and their impact on sphingolipid levels or enzymes of the sphingolipid pathway. HSPG: Heparan sulfate proteoglycane; aSMase: acid sphingomyelinase; CRP: ceramide rich platform; PC-PLC: phosphatidylcholine cholinephosphohydrolase; DAG: diacyl glycerides; IP3: inositol triphosphate; SREC-I: Scavenger receptor expressed by endothelial cells I; CEACAM: arcinoembryonic antigen-related cell adhesion molecule.

## Data Availability

Not applicable.

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
