# Peer review of "A Comprehensive Review on the Interplay between Neisseria spp. and Host Sphingolipid Metabolites"

_cells, 2021, doi:10.3390/cells10113201_

Round 1
Reviewer 1 Report
This is indeed a comprehensive review on the relationship between pathogenic Neisseria and sphingolipids produced by the host, covering literature since 1990th up to now. Human sphingolipids have long been implicated in the infection of pathogenic Neisseria. The recent progress on the metabolism and functional regulation of sphingolipids, technologies for labeling and imaging sphingolipids in live cells, and Neisseria pathogenesis provide critical stepping stones to the path of mechanistic understanding of Neisseria-host interactions through sphingolipids. Thus, such a review is timely and significant for the field. The review was well written, logically organized, and covered a large number of published papers. The following are suggestions for further improving the manuscript and a few grammatical corrections.
Specific comments:
- Sphingomyelinase activation and translocation are central points of the discussion in this manuscript. However, sphingomyelinase activation and translocation are not well defined here. Does sphingomyelinase activation mean turning on the enzymatic activity? The next question is the mechanisms by which sphingomyelinase enzymatic activity is turned on, for example, by proteolytic cleavage at low pH in late endosomes and lysosomes, phosphorylation by signaling-activated kinases, or binding of sphingomyelin in the membrane. Radiation (line 103), oxidation (line 105), and PC-PLC cannot directly activate aSMase and have to go through one of the mechanisms. Mixing them together appears to be confusing.
- Sphingomyelinase translocation also should be defined better, translocating from where to where. aSMases and nSMases distribute differently in human cells. Does translocation mean translocating to the plasma membrane? What triggers their translocation, and do aSMases and nSMases have the same triggers? Does translocation is one of the activation mechanisms? If aSMases translocate from lysosomes to the plasma membrane, will increased pH inactivate their enzymatic activities? These should be discussed, pointing out what is known and unknown.
- Readers will greatly appreciate it if the authors can discuss and speculate how the location and general cellular functions of sphingomyelinases relate to Neisseria pathogenesis. One puzzle is if the products of sphingomyelinases reduce gonococcal intracellular survival, why do gonococci activate sphingomyelinases.
- There is no discussion on the role of sphingosine kinases in Neisseria pathogenesis. If there is no evidence for their involvement, the sphingosine kinases section may not be necessary.
- Could the authors provide domain diagrams of aSMases and nSMases, which may help readers understand activation and translocation processes?
- Could the authors provide example images for ceramide staining in bacteria-infected human cells?
Minor corrections:
- Line 102, delete “Apparently.”
- From reference #112, the reference all shift one up. #112 should be #113, #113 should be #114, etc.
- Mg2+ should be Mg2+.
- Line 132, put a comma after peroxide.
- Line 305, phosphoinosit should be phosphoinositol.
- Line 355, 7.8 ug ml-1 should be 7.8 ug/ml or per ml.
Author Response
Point-by-point reply:
Reviewer #1
Specific comments:
- Sphingomyelinase activation and translocation are central points of the discussion in this manuscript. However, sphingomyelinase activation and translocation are not well defined here. Does sphingomyelinase activation mean turning on the enzymatic activity? The next question is the mechanisms by which sphingomyelinase enzymatic activity is turned on, for example, by proteolytic cleavage at low pH in late endosomes and lysosomes, phosphorylation by signaling-activated kinases, or binding of sphingomyelin in the membrane. Radiation (line 103), oxidation (line 105), and PC-PLC cannot directly activate aSMase and have to go through one of the mechanisms. Mixing them together appears to be confusing.
We are grateful for the careful comments of reviewer#1. As requested by the reviewer we incorporated a more detailed description and definition regarding the aspects of sphingomyelinase activation and sphingomyelinase translocation. The corresponding parts were rewritten and the appropriate literature is cited.
- Sphingomyelinase translocation also should be defined better, translocating from where to where. aSMases and nSMases distribute differently in human cells. Does translocation mean translocating to the plasma membrane? What triggers their translocation, and do aSMases and nSMases have the same triggers? Does translocation is one of the activation mechanisms? If aSMases translocate from lysosomes to the plasma membrane, will increased pH inactivate their enzymatic activities? These should be discussed, pointing out what is known and unknown.
We agree with the reviewers comment and a more detailed definition of sphingomyelinase translocation was included.
- Readers will greatly appreciate it if the authors can discuss and speculate how the location and general cellular functions of sphingomyelinases relate to Neisseria pathogenesis. One puzzle is if the products of sphingomyelinases reduce gonococcal intracellular survival, why do gonococci activate sphingomyelinases.
Indeed, this is an interesting aspect. Even if sphingosine is not a direct metabolite of the sphingomyelinase, the increased ceramide levels should be further metabolized to sphingosine. To date, it is not clear whether gonococcal infection of the host cell leads to sphingosine accumulation, or whether the bacterium additionally activates sphingosine kinases, for example, to avoid toxic sphingosine levels. Due to the lack of evidence for enzyme activation downstream of the sphingomyelinase, we would like to avoid including a discussion on this topic.
- There is no discussion on the role of sphingosine kinases in Neisseria pathogenesis. If there is no evidence for their involvement, the sphingosine kinases section may not be necessary.
The reviewer is right: although there is no clear direct role of sphingosine kinases in Neisseria pathogenesis, interference with sphingosine kinase activity results in increased gonococcal killing as outlined in the manuscript. In addition, we have obtained preliminary data clearly demonstrating a significant increase of sphingosine kinase activity in brain endothelial cells in response to N. meningitidis infection. We have inserted a corresponding sentence in the manuscript and pointed out that these data have not yet been published. However, we would like to share these data confidentially with the reviewer.
- Could the authors provide domain diagrams of aSMases and nSMases, which may help readers understand activation and translocation processes?
We agree with the reviewer’s comment. Instead of including an additional diagram we would like to refer to the clarification of these aspects in the main manuscript text as requested regarding reviewer’s comments 1 and 2.
- Could the authors provide example images for ceramide staining in bacteria-infected human cells?
As requested by the reviewer an additional example image of a ceramide staining in meningococcal infected human cells (now Figure 2) was included in the revised version of the manuscript.
Minor corrections:
- Line 102, delete “Apparently.”
In the revised manuscript, the term “apparently” has been removed.
- From reference #112, the reference all shift one up. #112 should be #113, #113 should be #114, etc.
We apologize for this error and have corrected the numbering of the references in the revised version of the manuscript
- Mg2+ should be Mg2+.
In the revised manuscript, Mg2+ was changed to Mg2+.
- Line 132, put a comma after peroxide.
A comma was added after at this point.
- Line 305, phosphoinosit should be phosphoinositol.
We apologize for this typo and corrected it in the revised version of the manuscript.
- Line 355, 7.8 ug ml-1 should be 7.8 ug/ml or per ml.
In the revised manuscript, µg ml-1 was changed to µg/ml.
Reviewer 2 Report
The manuscript is well written and interesting. The authors are experts in their respective fields. While a recent review on sphingolipids and pathogens was recently published (pmid 34408731 ), the present manuscript deals with Neisseria pathogens with the addition of two interesting paragraphs on antimicrobial activity and visualization. I have no major comments.
A number of references (around ref #100) are shifted by one reference.
Line 318 the manuscript abruptly switches from eukaryotic sphingolipids to bacterial sphingolipids. I suggest adding a few lines to introduce this part.
Line 273: two spaces between chronic and pelvic
Line 323: pportunity instead of opportunity.
Author Response
Point-by-point reply:
Reviewer #2
- A number of references (around ref #100) are shifted by one reference.
We apologize for this error and corrected the numbering of the references in the revised manuscript.
- Line 273: two spaces between chronic and pelvic
Unnecessary space was removed.
- Line 323: pportunity instead of opportunity.
We corrected the typing error in the revised manuscript
- Line 318 the manuscript abruptly switches from eukaryotic sphingolipids to bacterial sphingolipids. I suggest adding a few lines to introduce this part.
In the revised version of the manuscript, we have included a brief introduction to the topic.